# Xanthatin Alleviates LPS-Induced Inflammatory Response in RAW264.7 Macrophages by Inhibiting NF-κB, MAPK and STATs Activation

**DOI:** 10.3390/molecules27144603

**Published:** 2022-07-19

**Authors:** Yuanqi Liu, Wenyu Chen, Fang Zheng, Huanan Yu, Kun Wei

**Affiliations:** School of Bioscience and Bioengineering, South China University of Technology, Guangzhou 510006, China; yuanqi.l@foxmail.com (Y.L.); dcwy11@163.com (W.C.); zhengfang6666@163.com (F.Z.); yuhuananl@163.com (H.Y.)

**Keywords:** xanthatin, RAW 264.7, anti-inflammatory, ROS, NF-κB, MAPK, STAT3

## Abstract

Xanthatin (XT) is a sesquiterpene lactone isolated from the Chinese herb Xanthium, which belongs to the Asteraceae family. In this study, we developed an inflammation model via stimulating macrophage cell line (RAW 264.7 cells) with lipopolysaccharide (LPS), which was applied to assess the anti-inflammatory effect and probable mechanisms of xanthatin. When compared with the only LPS-induced group, cells that were pretreated with xanthatin were found to decrease the amount of nitric oxide (NO), reactive oxygen species (ROS) and associated pro-inflammatory factors (TNF-α, IL-1β and IL-6), and downregulate the mRNA expression of iNOS, COX-2, TNF-α, IL-1β, and IL-6. Interestingly, phosphorylated levels of related proteins (STAT3, ERK1/2, SAPK/JNK, IκBα, p65) were notably increased only with the LPS-activated cells, while the expression of these could be reverted by pre-treatment with xanthatin in a dose-dependent way. Meanwhile, xanthatin was also found to block NF-κB p65 from translocating into the nucleus and activating inflammatory gene transcription. Collectively, these results demonstrated that xanthatin suppresses the inflammatory effects through downregulating the nuclear factor kappa-B (NF-κB), mitogen-activated protein kinase (MAPK) and signal transducer and activator of transcription (STATs) signaling pathways. Taken together, xanthatin possesses the potential to act as a good anti-inflammatory medication candidate.

## 1. Introduction

Inflammation is an organism’s usual self-protective and tightly controlled strategy for avoiding harmful microbe invasion and for repairing tissues [1,2]. It is a part of the body’s complicated bio-responses to internal and external stimuli induced by different pathogens [3,4]. In general, the pre-inflammatory phase is when the body’s immune system is functioning and is a necessary stage for the body to repair itself. However, prolonged, persistent or excessive inflammation might cause wound healing to be delayed. In severe circumstances, it can become a chronic condition, such as rheumatoid arthritis, diabetes mellitus, asthma and metabolic syndromes [5,6]. Additionally, inflammation has long been known as a symptom and trigger of cancer [7]. Currently, anti-inflammatory medications can be divided into two main categories: steroids and non-steroids [8]. However, these medicines are not fully effective therapeutically and are associated with a number of side effects [9,10,11,12,13]. Therefore, more effective anti-inflammatory drugs with low side effects should be explored and discovered [12].

Lipopolysaccharide (LPS) is a key lipid constituent identified in the external membranes of gram-negative bacteria that triggers macrophages to establish models of inflammation [14,15]. Macrophages arrive at the site of damage and induce the production of a number of inflammatory mediators and factors, such as nitric oxide synthase (iNOS), cyclooxygenase-2 (COX-2), TNF-α, IL-1β and IL-6), thus mobilizing more immune cells and promoting the proliferation of keratin-forming cells, fibroblasts, and epithelial cells [16,17]. A family of Toll-like receptors (TLRs), which expressed in the outer cell membrane, plays a vital function in the immune system [18]. Toll-like receptor 4, the most well-studied protein expressed by immune cells, acts as a receptor for LPS to initiate the immunological response [19]. The Janus kinase-signal transducer and activator of transcription (JAK-STAT), nuclear factor kappa-B (NF-κB) and mitogen-activated protein kinase (MAPK) pathways are linked with immunology, cell growth and differentiation and cellular stress responses. When cells were subjected to unfavorable stimulatory conditions, these pathways were discovered to minimize inflammatory harm by controlling the expression of pro-inflammatory genes [20,21,22].

In China and Vietnam, *Xanthium strumarium L.* (Asteraceae) is a well-known traditional herbal medicine [23,24]. Traditional Chinese medicine has utilized the fruits and roots of the plant as an anti-inflammatory herb to treat ailments such as nasal sinusitis and arthritis [25]. The xanthium genus contains many bioactive ingredients, including sesquiterpenes, monoterpene glucosides, essential oils, etc. [26,27,28]. In previous studies, the active ingredients in *X. strumarium* have been demonstrated to possess antibacterial, anticancer, antioxidant, and anti-inflammatory activities [29,30,31,32,33]. From the extracts of *X. strumarium* roots, some phenolic and flavonoid substances exhibited antioxidant and antitumor activities [34]. Caffeoylquinic acid, which is found in the fruits of *X. strumarium*, has anti-inflammatory and analgesic properties [35]. In human breast cancer cells, xanthatin (XT) has been demonstrated to block the activator of transcription 3 (STAT3) and NF-κB signaling pathways [25]. In addition, in asthmatic mice and corneal alkali burn models, xanthatin exerted anti-inflammatory activities via blocking STAT3, NF-κB or PI3K/Akt pathways and promoted apoptosis [25,29,36].

To the best of our knowledge, the direct effects of xanthatin on anti-inflammatory effects have not been comprehensively investigated. In this paper, we used LPS stimulation of macrophage cell line (RAW 264.7 cells) to create an inflammatory model that was used to evaluate the anti-inflammatory effectiveness of xanthatin and explore its promising mechanisms.

## 2. Results

### 2.1. Cell Viability of Xanthatin

The CCK-8 Cell Proliferation and Cytotoxicity Assay Kits (CCK-8 kits) were used to examine the toxic effect of xanthatin in the absence or presence of LPS (1 μg/mL) on RAW 264.7 cells (Figure 1). First, cells were pre-treated with different concentrations (0–50 μM) of xanthatin for 24 h to assess the toxicity. Also, cells were pre-treated with different concentrations (0–50 μM) of xanthatin in the absence or presence of LPS for 24 h. The concentrations (0–12.5 μM) of xanthatin exhibited no difference in cytotoxicity. The cell viability of RAW 264.7 was not affected in the presence of 1 μg/mL LPS. Therefore, the data indicated that cell viability was not affected by xanthatin (0–12.5 μM) and LPS (1 μg/mL). Furthermore, changes were observed in the morphology of the RAW 264.7 cells in the blank group, LPS group, and different concentrations of the xanthatin treated group. LPS induction caused changes in the cellular morphologies, which also implied the successful establishment of the inflammation model. However, pretreatment with xanthatin suppressed the change of irregular cell morphology, indicating that it could inhibit cell inflammation to some extent.

### 2.2. Effects of Xanthatin on LPS-Stimulated on NO Production 

To characterize the effect of xanthatin on LPS-induced NO release, we employed a Griess reagent. When cells were activated by LPS, the release of NO was increased significantly, which implied that the inflammatory model was successful. The results showed that the amount of NO reduced dramatically in a dose-dependent way, compared to the LPS-induced group. The NO content inhibition rates were 16.7%, 29.7%, 53.4% and 76.8% at 0.78, 1.56, 3.125 and 6.25 μM concentrations of xanthatin, respectively (Figure 2).

### 2.3. Effects of Xanthatin on LPS-Induced Pro-Inflammatory Mediator Expression

Enzyme linked immunosorbent assay kits (ELISA kits) were utilized to detect pro-inflammatory mediators (TNF-α, IL-1β and IL-6) release (Figure 3). Cells generated a number of cytokines and mediators only in the LPS-stimulated group. The results showed that TNF-α, IL-1β and IL-6 production levels were increased in the culture supernatants of LPS-induced cells. These changes were notably attenuated in a dose-dependent way when cells were pre-treated with xanthatin. Taken together, the above results implied that xanthatin can suppress the inflammatory stimulus response by reducing TNF-α, IL-1β and IL-6 levels.

### 2.4. Effects of Xanthatin on LPS-Induced iNOS, COX-2, TNF-α, IL-1β, and IL-6 mRNA Levels

The production of pro-inflammatory cytokines is the most direct manifestation of the presence of inflammation. To fully understand the anti-inflammatory function of xanthatin in LPS-induced cells, we investigated the mRNA levels of iNOS, COX-2, TNF-α, IL-1β and IL-6 using quantitative reverse transcriptase-polymerase chain reaction (qRT-PCR). Xanthatin can effectively suppress the mRNA levels of iNOS and COX-2 in a dose-dependent way, as compared to the LPS-stimulated group. TNF-α, IL-1β and IL-6 levels followed the same trend as inflammatory mediators (Figure 4).

### 2.5. Effects of Xanthatin on LPS-Stimulated on the Reactive Oxygen Species (ROS) Content

Inflammatory stimulation induces large amounts of ROS production by macrophages. ROS can mediate related pathways to produce pro-inflammatory responses and prolong the duration of inflammation. The amount of ROS released was considerably enhanced under LPS-induced RAW 264.7 cells, while the intracellular ROS decreased in a dose-dependent way with a xanthatin pre-treatment (Figure 5). These results indicated that xanthatin works to reduce ROS content and thus inhibits inflammation development and transmission. 

### 2.6. Effects of Xanthatin on LPS-Induced on the iNOS and COX-2 Expression

INOS and COX-2 proteins regulate the production of NO and PGE2 respectively. To understand the inhibitory effect of xanthatin on LPS-stimulated in RAW 264.7 cells, we used Western blotting to explore the expression of iNOS and COX-2 (Figure 6). The result showed that the levels of iNOS and COX-2 were upregulated in response to the LPS-induced group. When compared to the group that was pre-treated with xanthatin, both of their mRNA were downregulated in a dose-dependent way.

### 2.7. Effects of Xanthatin on LPS-Induced on the NF-κB, MAPK and STATs Signaling Pathways

To fully explore the molecular mechanism of the inhibitory inflammation activity of xanthatin on LPS induction in RAW 264.7 cells, we conducted Western blotting to analyze the expression of NF-κB, STATs and MAPK pathway-related proteins (p-IκBα, IκBα, p-p65, p-STAT3, STAT3, p-ERK1/2, ERK1/2, p-SAPK/JNK and SAPK/JNK). There are relevant cellular signaling pathways in the inflammatory response induced by LPS. Among them, cells were pre-treated with xanthatin or only the LPS-induced group, which had little influence on the non-phosphorylated proteins. However, phosphorylated proteins were markedly increased only with the LPS-induced group. When cells were pre-treated with xanthatin, these alterations were reversed in a dose-dependent way (Figure 7). Above all, cells pre-treated with xanthatin could suppress the inflammatory cytokines by blocking the phosphorylation of NF-κB, STATs and MAPK pathway proteins.

### 2.8. Effects of Xanthatin on LPS-Induced on the NF-κB p65 Nuclear Translocation

Translocation of NF-κB p65 from the cytoplasm into the nucleus triggers an inflammatory response. To investigate whether xanthatin has an inhibitory effect on LPS-activated NF-κB p65 nuclear translocation, it was observed by laser confocal scanning electron microscopy. As shown in Figure 8, under normal circumstances, NF-κB p65 (red) was largely distributed in the cytoplasm. NF-κB p65 was transposed to the nucleus with LPS-activation, while the red color in the nucleus was significantly diminished by co-culture with xanthatin, indicating that pre-treatment with xanthatin could partially reverse the phenomenon of nuclear translocation.

## 3. Discussion

Inflammation is a normal response to a complex pathological process, which is the body’s defense response to injurious stimuli or pathogenic components [37]. However, excessive and prolonged inflammation is one of the reasons for the initiation and worsening of a wide range of disorders, for example, autoimmune diseases and chronic inflammation [15]. Therefore, controlling the timing and progression of inflammation is a key point in healing. The present study focused on the anti-inflammatory properties of xanthatin on LPS-induced RAW 264.7 cells, including the expression of inflammatory factors and their associated signaling pathways. Xanthatin showed no toxic effects on RAW 264.7 cells at concentrations below 12.5 μM. RAW 264.7 cell viability was unaffected by xanthatin with doses (0–12.5 μM) or LPS (1 μg/mL). To assess the anti-inflammatory effectiveness, various concentrations of xanthatin were utilized.

In the inflammatory response, macrophages are the most responsive innate immune cells, acting as both reactors and sensors to modulate inflammatory and immunological responses. When activated by LPS, macrophages secrete pro-inflammatory cytokines (TNF-α, IL-1β and IL-6) and NO to enhance the immune response. To amplify and deliver the immune response, RAW 264.7 cells greatly boosted the release and transcription of TNF-α, IL-1β, IL-6, iNOS and COX-2 after being LPS-induced. Pre-treatment with xanthatin reversed this phenomenon. These findings imply that xanthatin may prevent the release of LPS-induced pro-inflammatory factors, resulting in protective and anti-inflammatory effects.

TNF-α binds to the receptor (TNFR), causing it to rapidly synthesize and secrete a variety of cytokines and chemokines, which were capable of transcriptionally upregulating a series of inflammatory gene expression events, all of which contribute to inflammation [38,39]. IL-1β is an important pro-inflammatory cytokine that is produced by a variety of cell types and is involved in the chemotaxis of immune cells such as neutrophils and macrophages [40,41]. In the inflammatory response, IL-6 is a biomarker with pro-inflammatory and anti-inflammatory properties [41,42]. It can operate as a key element in attracting T cells, while also reducing neutrophils to reach the wound site [43,44]. In this study, RAW 264.7 cells were followed by an increase in pro-inflammatory cytokine expression at the levels of mRNA and proteins after being LPS-induced. However, pre-treatment with xanthatin decreased their production. From these results, it is clear that xanthatin can achieve a state of balance by suppressing the massive production of pro-inflammatory factors, thus exerting the maximum immune effect.

NO and PGE2 are critical messenger molecules and indicators of inflammation [45,46]. NO is a free radical formed by the iNOS family [47]. INOS, a part of M1 macrophages, plays a role in the inflammatory response, particularly in reaction to LPS [48]. The high expression of iNOS leads to NO generation over the threshold, which slows wound healing. In the creation of PG, COX (COX-1, COX-2) plays a main role, and COX-2 controls PGE2 expression to reduce inflammation [49,50]. Xanthatin reduces NO generation by downregulating iNOS expression in terms of mRNA and protein. Meanwhile, transcription and expression of the COX-2 gene were affected by the pre-treatment of xanthatin, showing concentration-dependent suppression. These findings show that xanthatin inhibits the expression of similar synthases, hence reducing the generation of inflammatory mediators.

As intracellular pro-inflammatory signaling molecules, ROS have the ability to regulate the immune response [51]. ROS plays a dual role in inflammation [52,53]. It can operate as a redox signal to control the length of time that inflammation lasts [54]. The formation of inflammatory vesicles is exacerbated by the accumulation of large amounts of ROS in injured tissue [55]. Furthermore, ROS can mediate pro-inflammatory reactions via the NF-κB and STATs pathways and stimulate the secretion of pro-inflammatory cytokines [56,57]. In this investigation, the cells that were pre-treated with xanthatin dramatically decreased ROS generation after LPS stimulation. Xanthatin reduced tissue damage and inflammation by removing excess ROS. Further research is needed to assess the possibility that xanthatin reduces ROS levels in relation to inhibition of the NF-κB pathway.

Keratinocytes in the epidermis are the outermost layer of protection and generate various pattern recognition receptors (PRRs) [58,59]. They mainly recognize danger signals with specific small molecular motifs, which are called pathogen-associated molecular patterns (PAMPs) [59]. Some tissue fragments, pathogens, and LPS are recognized by PRRs. Toll-like receptor 4 (TLR4) recognize the chemical structure of LPS and trigger an inflammatory response to clear the foreign substance from the body [60]. Neutrophils and macrophages are recruited to reach the site. Activated macrophages produce pro-inflammatory effects and secrete multiple signaling factors to activate relevant signaling pathways [51,52]. The STATs pathway, which regulates the developmental and internal homeostatic balance of the organism through a series of signaling cascades, is involved in the immune response [53,54]. STAT3 is a characteristic inflammatory signaling molecule in the STAT protein family [51]. Activated STATs translocate to the nucleus, in which they affect gene transcription of related factors and enzymes (like iNOS and COX-2) [55]. ROS played an important role in mediating STAT to regulate the expression of inflammatory genes. STAT phosphorylation can also be regulated by MAPK and NF-κB pathways. In the above study, p-STAT3 expression was greatly elevated in RAW 264.7 macrophages with LPS-induction while lowered by pre-treatment with xanthatin in a concentration-related manner, suggesting that STAT3 may serve as a target for xanthatin to inhibit inflammation. This study suggests that xanthatin is capable of producing anti-inflammatory effects via the STAT pathway, possibly due to an upstream pathway or because of ROS mediation.

NF-κB and MAPK are critical cellular signaling pathways in the inflammatory response promoted by LPS [56]. STAT3 and NF-κB are central transcription factors in immunity in response to LPS stimulation [57]. NF-κB and MAPK play a role as regulators in the process of inflammatory signaling and are involved in the transcription of various pro-inflammatory mediators and cytokines [21,61]. STAT3 and ROS also affect NF-κB mediated pro-inflammatory responses [62,63]. With abnormal stimulation, NF-κB family member p65 is translocated from the dimer to the nucleus, where it triggers the development of TNF-a, IL-1β and IL-6 by binding to a number of target genes [64]. P65 in the cytoplasm dissociates from the heterodimer and translocates to the nucleus to combine with pro-inflammatory genes when a series of cascade reactions are triggered by inflammation [65]. The MAPK pathway is involved in the regulation of inflammatory cytokines and chemokines (iNOS and COX-2), and also in the activation of NF-κB [46,63,66]. However, not all target proteins in the pathway play the same role. In the current investigation, xanthatin was able to effectively inhibit the phosphorylation of ERK1/2 and JNK. The anti-inflammatory effect of xanthatin is through the inhibition of NF-κB and MAPK signal transduction, but its most direct target is not yet clear, and more exploration is needed in the future.

The inflammatory process is mediated by a variety of inflammatory mediators and cytokines that are major causes of inflammatory illnesses [67]. Therefore, blocking the production of these pro-inflammatory mediators may be an effective treatment strategy [68]. LPS stimulates TLR4, which activates NF-κB and MAPK to act as a signaling cascade to regulate the transcription of target genes encoding pro-inflammatory cytokines, chemokines, and inducible enzymes [69,70,71,72]. ROS are signaling molecules that can activate various signaling pathways, such as MAPK and STATs pathways [73,74]. Furthermore, STAT3 interacts with NF-κB and MAPK pathways to inhibit LPS-induced NO, TNF-α, IL-1β and IL-6 transcription and expression [75,76]. Xanthatin down-regulated the transcription of iNOS, COX-2, TNF-α, IL-6 and IL-1β mRNA induced by LPS stimulation. In addition, at the protein level, it also inhibited the expression of iNOS, COX-2 and pathway-related proteins. These results suggest that xanthatin inhibits the transcription and protein translation of the relevant pro-inflammatory genes and proteins to some extent. Furthermore, xanthatin exerts its anti-inflammatory effects through the interaction between ROS signaling molecules and NF-κB, MAPK and STATs signaling pathways to modulate inflammatory factors and mediators. In this study, the positive anti-inflammatory effect of xanthatin in *Xanthium sibiricum Patr.* has been explored. In addition, it has other active ingredients which have many potent medicinal values and biological activities such as antibacterial, anti-inflammatory, analgesic, antitumor, antioxidant and hypoglycemic effects [26]. These rich functional active ingredients mean that Xanthium has potential as a source of novel nutraceuticals.

## 4. Materials and Methods

### 4.1. Materials

Xanthatin (XT) was purchased from Chengdu Herbpurify CO., LTD. LPS (*E.coli* 0111:B4) and dimethyl sulfoxide (DMSO) was purchased from Sigma-Aldrich (St. Louis, MO, USA). Dulbecco’s Modified Eagle Medium (DMEM), fetal bovine serum (FBS), and antibiotic-antimycotic solution were purchased from Gibco (ThermoFisher, Gaisburg, MD, USA). CCK-8 kits were purchased from (GLPBIO). Enzyme-linked immunosorbent assay (ELISA) kits for TNF-α, IL-6, IL-1β were purchased from BOSTER Biological Technology (Wuhan, China). The primary rabbit monoclonal antibodies against STAT3, p-STAT3, SAPK/JNK, p- SAPK/JNK, NF-κB p65, p-NF-κB p65, ERK1/2, p-ERK1/2, IκBα, p-IκBα were purchased from Cell Signaling Technology (Danvers, MA, USA). The primary rabbit monoclonal antibodies against iNOS and COX-2 were purchased from Affinity biosciences OH (USA).

### 4.2. Cell Culture

A RAW 264.7 murine macrophage cell line was purchased from the National Collection of Authenticated Cell Cultures (Shanghai, China), and cultured in DMEM supplemented with 10% FBS, 100 U/mL penicillin and 100 mg/mL streptomycin, in a humidified incubator with 5% CO_2_ at 37 °C.

### 4.3. Cell Viability Assay

The CCK-8 kits were applied to evaluate cell viability. Cells were seeded and pre-treated with various concentrations of xanthatin in the absence or presence of LPS for 24 h in 96-well plates. Then, the morphological changes of RAW 264.7 cells were observed by inverted fluorescence microscopy.10 μL CCK-8 was added and incubation occurred for 1 h at 37 °C. A multi-technology microplate reader was used to detect the absorbance at 490 nm. At least three replicates were included in each sample.

### 4.4. Measurement of Nitric Oxide (NO)

The Griess reagents (Beyotime, Shanghai, China) were applied to test the release of NO. Cells were seeded and pre-treated with various concentrations of xanthatin in the absence or presence of LPS for 24 h. Then, the supernatant medium of 50 μL in each group was transferred and mixed with the same volume of Griess I and II reagent (50 μL) in 96-well plate. A modular multi-technology microplate reader was used to detect the absorbance at 540 nm.

### 4.5. ELISA

The expression of cytokines was tested by the mouse ELISA kits (TNF-α, IL-1β and IL-6). Cells were seeded and pre-treated with various concentrations of xanthatin in the absence or presence of LPS for 24 h in 12-well plates. Then each supernatant was collected in a tube. The content of each factor is identified following the instructions of the ELISA kit. 

### 4.6. Quantitative Reverse Transcriptase-Polymerase Chain Reaction (qRT-PCR)

Cells were pre-treated with various concentrations of xanthatin in the absence or presence of LPS for 24 h in 6-well plates. Extraction of total cellular RNA from cells was achieved by using a Trizol reagent. RNA in each sample was reversed to single-stranded cDNA according to the following recipe and procedure: 20 uL of final reverse transcription reaction solution (4 μL of 5X buffer,1 μL of Random Primers, 2 μL of Enzyme Mix, RNA 1.5 g). 25 °C, 5 min, 42 °C, 30 min, 85 °C, 5 min. The relevant primers and their sequences are listed in Table 1. QPCR was conducted according to the standard thermal cycling method [77]. The mRNA levels were normalized to β-Actin. At least three replicates were performed on each sample for independence. 

### 4.7. Reactive Oxygen Species (ROS) Measurement

Cells were seeded and pre-treated with different concentrations of xanthatin with or without LPS for 24 h, after washing cells with cold-PBS and staining it by DCFH-DA at 10 μM for 0.5 h. Performed experiments according to the kit’s instructions. Collected each cell pellet, centrifuged, and resuspended with PBS. Used a flow cytometer (BD FACS Aria) to detect the ROS in cells.

### 4.8. Western Blot Analysis

Cells were seeded and pre-treated with various doses of xanthatin with or without LPS for 24 h in 6-well plates. The RIPA lysis buffer was applied to each cultivated cell after it was washed twice with cold PBS. Later, the RIPA lysis buffer was collected and centrifugated. A BCA protein assay kit was used to assess each concentration of the protein extraction. Protein samples (30 μg) were electrophoresed on 8–12% SDS-PAGE gels and transferred to PVDF membranes for 1–4 h at 100 V. The membranes were blocked in 5% non-fat dry milk in TBST for 1 h with slight shaking. Later, the PVDF membranes were washed three times with TBST and then incubated overnight at 4 °C with primary rabbit monoclonal antibodies. Washed and incubated for 90 min at room temperature with HRP-conjugated antibodies. Protein signals were visualized using a chemiluminescence kit (Millipore Corporation, Billerica, MA, USA) with the ChemiDoc-It Imaging System (UVP, Upland, CA, USA).

### 4.9. The NF-κB p65 Nuclear Translocation Immunofluorescence Staining

The NF-κB p65 translocation was investigated using immunofluorescence microscopy. Cells were seeded and pre-treated with different doses of xanthatin with or without LPS for 24 h in 12-well plates. The operation primarily refers to our group’s testing procedure [77]. Finally, the crawls were visualized by a laser scanning confocal microscope (Nikon Ti-EA1, Japan).

### 4.10. Statistical Analysis

Experimental graphs were obtained with three replicate experiments, and one-way ANOVA and an independent t-test were utilized to identify the results’ statistical significance. A noticeable difference was defined as *p* < 0.01. Data was analyzed using Origin 2018.

## 5. Conclusions

In this study, the efficacy and mechanism of xanthatin for the inflammation model established by LPS were studied. Xanthatin exhibits potent anti-inflammatory effects. In LPS-stimulated RAW 264.7 cells, xanthatin can inhibit the production of NO, the expression of IL-1β, IL-6 and TNF-α, and the transcription of IL-1β, IL-6, TNF-α, iNOS and COX2. Concurrently, the upstream signaling proteins were evaluated. Xanthatin reduced the expression levels of iNOS and COX-2 proteins and also inhibited the expression levels of IκBα, p65, STAT3, ERK1/2 and SAPK/JNK phosphorylated proteins. In addition, xanthatin can lower ROS content to reduce cell damage and pathway activation. These changes could suggest that xanthatin inhibits the occurrence of inflammation through the interaction of NF-κB, MAPK and STATs pathways. As a whole, xanthatin improves the inflammatory response induced in RAW 264.7 macrophages when LPS-activated, which offers us ideas for understanding and developing *Xanthium L.* plants as a treatment for inflammation.

## Figures and Tables

**Figure 1 molecules-27-04603-f001:**
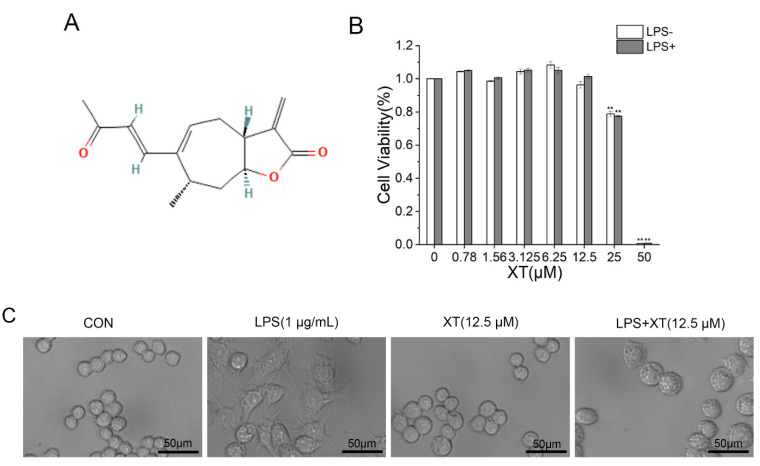
Cell viability of xanthatin. (**A**) The chemical structure of xanthatin. (**B**) Cytotoxicity of xanthatin at various concentrations (0–50 μM) in the presence of LPS on RAW 264.7 cells, followed by the CCK-8 kits. (**C**) Microscopy images of cellular morphology. Scale bar = 50 μm. Data is shown as means ± SDs (n = 3). #: significantly different compared to the control (CON) group, **: *p* < 0.01 as compared to the LPS-induced group.

**Figure 2 molecules-27-04603-f002:**
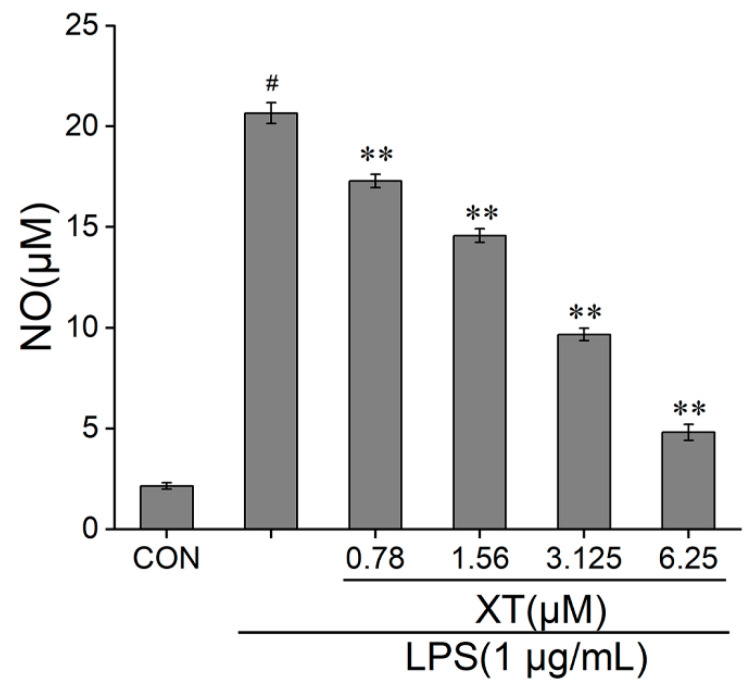
Effects of xanthatin on NO release in RAW 264.7 cells. Cells were pre-treated with different concentrations of xanthatin for 3 h and incubated with LPS for another 21 h. The NO content in the culture supernatant were tested by a Griess reagent. The NO content was decreased with xanthatin in a dose-dependent way. Data is shown as means ± SDs (n = 3). #: significantly different compared to the control (CON) group, **: *p* < 0.01, as compared to the LPS-induced group.

**Figure 3 molecules-27-04603-f003:**
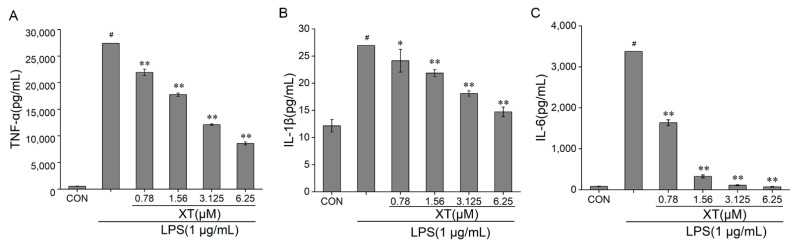
Effects of xanthatin on release of pro-inflammatory factors: TNF-α (**A**), IL-1β (**B**), IL-6 (**C**). When cells were pre-treated with xanthatin and activated by LPS, the concentration of TNF-α, IL-1β, IL-6 in the culture supernatants was tested using an ELISA kit assay. Data is shown as means ± SDs (n = 3). #: significantly different compared to the control (CON) group, *: *p* < 0.05, **: *p* < 0.01, as compared to the LPS-induced group.

**Figure 4 molecules-27-04603-f004:**
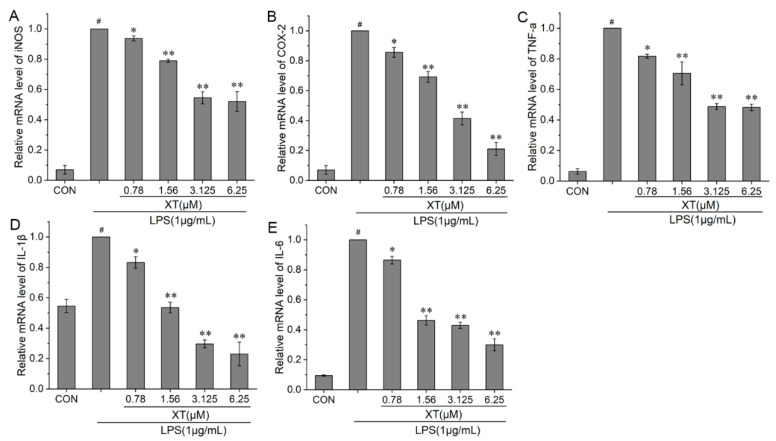
Effects of xanthatin on LPS-induced mRNA levels of iNOS (**A**), COX-2 (**B**), TNF-α (**C**), IL-1β (**D**) and IL-6 (**E**) in RAW 264.7 cells. When cells were pre-treated with xanthatin and activated by LPS, the mRNA levels of iNOS, COX-2, TNF-α, IL-1β and IL-6 was detected by qPCR. These results for the related mRNA expression levels were normalized with β-actin. Data is shown as means ± SDs (n = 3). #: significantly different compared to the control (CON) group, *: *p* < 0.05, **: *p* < 0.01, as compared to the LPS-induced group.

**Figure 5 molecules-27-04603-f005:**
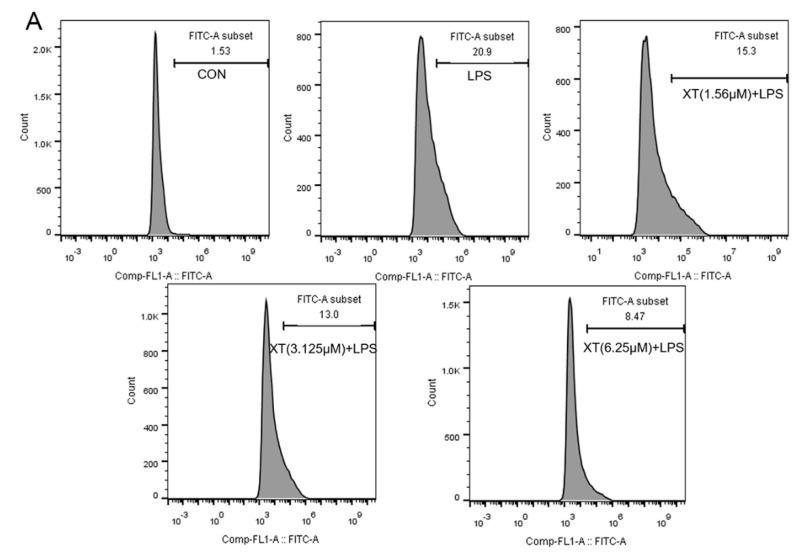
Effects of xanthatin on ROS release in RAW 264.7 cells. When cells were pre-treated with xanthatin and activated by LPS, we used flow cytometry to test the intracellular ROS generation. This graph shows the roles of xanthatin pre-treated and then LPS-induced on ROS formation. (**A**) Flow cytometry was processed using the BD Accuri C6 Plus software. (**B**) The ROS content was expressed as the percentage of cells with bright fluorescence. Data is shown as means ± SDs (n = 3). #: significantly different compared to the control (CON) group, **: *p* < 0.01, as compared to the LPS-induced group.

**Figure 6 molecules-27-04603-f006:**
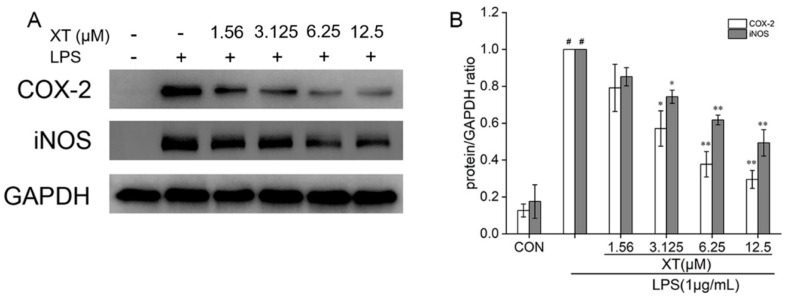
Effects of xanthatin on iNOS and COX-2 protein expression in RAW 264.7 cells. When cells were pre-treated with xanthatin and then activated by LPS, we clarified the protein of iNOS and COX-2 (**A**) by Western blotting. The relative expression grey value of each protein band is shown (**B**). GAPDH was considered as an internal reference. Data is shown as means ± SDs (n = 3). #: significantly different compared to the control (CON) group, *: *p* < 0.05, **: *p* < 0.01, as compared to the LPS-induced group.

**Figure 7 molecules-27-04603-f007:**
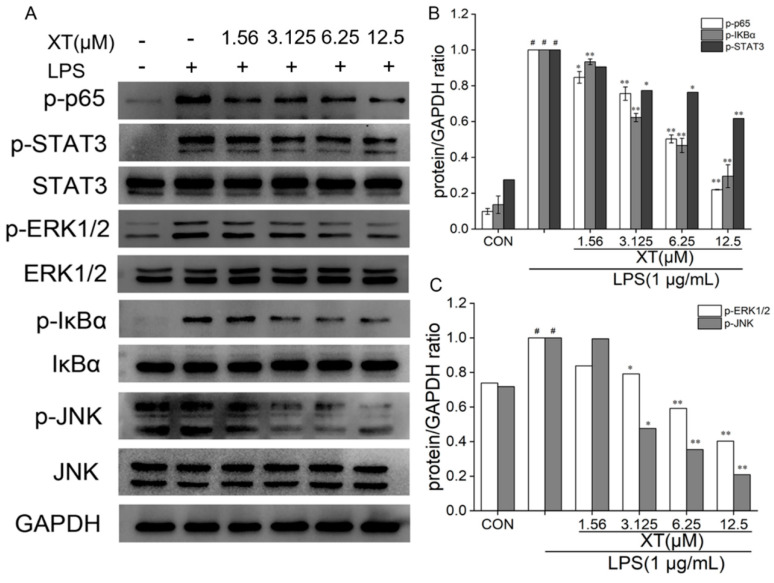
Effects of xanthatin on NF-κB, MAPK and STATs (**A**) protein expression in LPS-induced RAW 264.7 cells. The relative expression grey value of each protein band is shown (**B**,**C**). GAPDH was considered as an internal reference. #: significantly different compared to the control (CON) group, *: *p* < 0.05, **: *p* < 0.01, as compared to the LPS-induced group.

**Figure 8 molecules-27-04603-f008:**
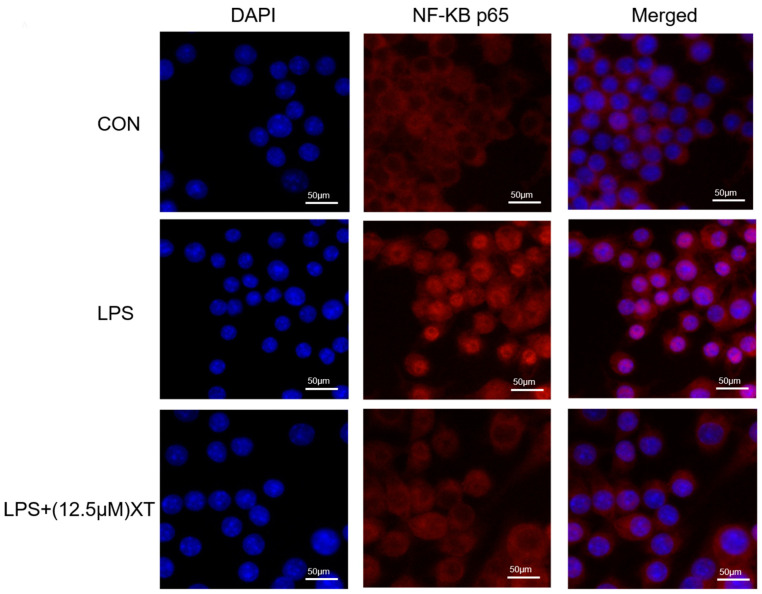
Effects of xanthatin on the NF-κB p65 nuclear translocation in LPS-induced RAW 264.7 cells. Scale bar = 50 μm. When cells were pre-treated with xanthatin and activated by LPS, we used it to evaluate the NF-κB p65 nuclear translocation by laser confocal scanning electron microscopy. NF-κB p65 antibodies (red) and DAPI (blue) were used to stain the cells.

**Table 1 molecules-27-04603-t001:** Reverse transcription polymerase chain reaction primers were utilized.

Name	Primer Sequence (5′-3′)	Product (bp)
iNOS	F: GCTATGGCCGCTTTGATGTGR: ACCTCCAGTAGCATGTTGGC	184
COX-2	F: TGAGTACCGCAAACGCTTCTR: ACGAGGTTTTTCCACCAGCA	148
TNF-α	F: GACGTGGAACTGGCAGAAGAR: ACTGATGAGAGGGAGGCCAT	192
IL-1β	F: AACCTTTGACCTGGGCTGTCR: AAGGTCCACGGGAAAGACAC	144
IL-6	F: ATCCAGTTGCCTTCTTGGGAR: GGTCTGTTGGGAGTGGTATCC	103
β-actin	F: AGGGAAATCGTGCGTGACATR: AACCGCTCGTTGCCAATAGT	149

## Data Availability

The data presented in this study are available on request from the corresponding author.

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
