# Peer review of "Xanthatin Alleviates LPS-Induced Inflammatory Response in RAW264.7 Macrophages by Inhibiting NF-κB, MAPK and STATs Activation"

_molecules, 2022, doi:10.3390/molecules27144603_

Round 1
Reviewer 1 Report
The paper prepared by Liu et al. is clear, well organized and well written. This manuscript deals with the study on anti-inflammatory effectiveness and mechanism of xanthatin (product originating from medicinal plant) used in the case of LPS stimulated inflammatory model using RAW 264.7 cells. Well-documented research results indicate that xanthatin has a potential anti-inflammatory effect.
This manuscript can be published in Molecules, however, before acceptance the authors should consider a couple of minor revision points:
Abbreviations and acronyms should be defined for the first time when they are used, throughout the abstract and again the text of manuscript. Thus, abbreviation LPS should be explained both in the Abstract and in the manuscript. Moreover, some terms should be shortly explained as they are not commonly known. There are: TLR receptors, RAW 264.7 cells, CCK-8 kits, ELISA kits, qPCR, ROS, PRRs.
Table 1 has not been cited in the text of manuscript. Additionally, Figure 9 mentioned in section 5. Conclusions, is missing.
References – no bibliographic data (volume or page numbers) for many publications, for example, refs. 27, 34, 35, 41, 70, 73-75, 77-78.
Author Response
Dear Referee:
We would first like to thank you very much for reviewing our manuscript. Those comments concerning our manuscript are constructive and helpful for revising and improving our manuscript. We have revised our manuscript according to your comments. All revisions to the manuscript have been marked in red and marked up using the “Track Changes” function in the paper.

Reviewer 2 Report
This is a very well-written manuscript and it merits publication. All parts of the paper are well written and the data are well presented by the authors.
I have a comment about figure 5 where at the current format, it's not very clear to the reader. I would suggest the authors to break this figure into two figures so the fonts will be bigger and it will be readable.
My second comment is on the discussion/conclusions sections, where I would invite the authors to discuss the potential of using Xanthium as a source of novel nutraceuticals.
Therefore, only a minor revision is needed.
Author Response

(The authors gave the same response as above.)
